# Epilepsy in Down Syndrome: A Highly Prevalent Comorbidity

**DOI:** 10.3390/jcm10132776

**Published:** 2021-06-24

**Authors:** Miren Altuna, Sandra Giménez, Juan Fortea

**Affiliations:** 1Sant Pau Memory Unit, Department of Neurology, Hospital de la Santa Creu i Sant Pau, Biomedical Research Institute Sant Pau, Universitat Autònoma de Barcelona, 08041 Barcelona, Spain; sgimenez@santpau.cat; 2Center of Biomedical Investigation Network for Neurodegenerative Diseases (CIBERNED), 28031 Madrid, Spain; 3Multidisciplinary Sleep Unit, Respiratory Department, Hospital de la Santa Creu i Sant Pau, 08041 Barcelona, Spain; 4Barcelona Down Medical Center, Fundació Catalana de Síndrome de Down, 08029 Barcelona, Spain

**Keywords:** down syndrome, aging, seizures, epilepsy, Alzheimer’s disease

## Abstract

Individuals with Down syndrome (DS) have an increased risk for epilepsy during the whole lifespan, but especially after age 40 years. The increase in the number of individuals with DS living into late middle age due to improved health care is resulting in an increase in epilepsy prevalence in this population. However, these epileptic seizures are probably underdiagnosed and inadequately treated. This late onset epilepsy is linked to the development of symptomatic Alzheimer’s disease (AD), which is the main comorbidity in adults with DS with a cumulative incidence of more than 90% of adults by the seventh decade. More than 50% of patients with DS and AD dementia will most likely develop epilepsy, which in this context has a specific clinical presentation in the form of generalized myoclonic epilepsy. This epilepsy, named late onset myoclonic epilepsy (LOMEDS) affects the quality of life, might be associated with worse cognitive and functional outcomes in patients with AD dementia and has an impact on mortality. This review aims to summarize the current knowledge about the clinical and electrophysiological characteristics, diagnosis and treatment of epileptic seizures in the DS population, with a special emphasis on LOMEDS. Raised awareness and a better understanding of epilepsy in DS from families, caregivers and clinicians could enable earlier diagnoses and better treatments for individuals with DS.

## 1. Introduction

Down syndrome (DS), caused by an extra chromosome 21, is the most frequent cause of intellectual disability of genetic origin [1,2,3,4,5,6]. Its prevalence ranges from 1/650 to 1/1000 births [7,8,9]. During the last decades the life expectancy of people of DS has significantly increased, and it is estimated that children with DS born in 2010 will now live up to 60–65 years [10]. Increased survival coupled with accelerated aging [8] has led to an increase in the prevalence of age-related diseases, most importantly Alzheimer’s disease (AD), which has a cumulative incidence >90% at age 65 years [4,8,11,12].

People with DS have a higher prevalence of epilepsy, compared to the general population, similarly to other forms of intellectual disability [1,13,14,15,16]. The prevalence of seizures at any age in DS is estimated to be between 8.1–26% (versus 1.5–5% in the general population) [16,17] and has a clear bimodal distribution [3,9,18,19,20,21,22,23,24,25]. The first peak/mode of incidence occurs in the early childhood (0–2 years) of life and the second in the 6th decade [9,17,26,27]. This latter peak incidence is closely related to symptomatic AD [2,17,28,29,30,31,32]. Due to its specific clinical presentation and characteristics, it has received a specific name, late onset myoclonic epilepsy in DS (LOMEDS). LOMEDS is a generalized myoclonic epilepsy with an important burden in patients and their families. It has an impact on the quality of life and on mortality and might have worse cognitive and functional outcomes in patients with AD dementia.

We will review the risk factors and clinical characteristics of epilepsy during childhood and adulthood in people with DS, with a special emphasis on LOMEDS. We aim to raise awareness in families, caregivers and clinicians to enable earlier diagnoses and better treatments for individuals with DS and epilepsy.

## 2. Methodology

We performed a literature review using PubMed and Web of Science (WOS), combining two independent but complementary search strategies with the following Mesh terms: “Down Syndrome”, “Intellectual Disability”, “Alzheimer Disease”, “Epilepsy”, “Seizures” and “Electroencephalography” (Figure 1). We did not apply any time restriction, and we included original articles and review articles with data from human subjects (exclusion of papers in animals only) written in English, Spanish or French available at 1st of May 2021. We selected articles with abstracts available in PubMed or WOS. After reading the titles and abstracts, those papers that met eligibility criteria were selected for full-text revision. Those papers specifically dealing with epileptic seizures and epilepsy in Down syndrome are included for this review.

The review is divided into four main sections: (1) epilepsy in early childhood (under 5 years old); (2) epilepsy between 5 years old and late adulthood; and (3) Epileptic seizures in late adulthood (≥40 years) and their relationship with symptomatic AD. In each main section we reviewed the available data on pathophysiology, clinical and electrophysiological characteristics. We finalize with (4) a section on treatment recommendations and future directions.

Some important terms from the current epilepsy nomenclature are reviewed in Table 1 [33,34].

## 3. Epilepsy in Early Childhood (0–5 Years)

Similarly to other forms of intellectual disability, children with DS have a higher prevalence of epilepsy than typically developing children [7,35,36]. Children with DS have several types of seizures and an increased risk of developing some specific epileptic syndromes. This risk typically occurs in children before the first year of life, but some seizures such as the febrile seizures (typically developing between 6 months and 5 years) or the Lennox-Gastaut syndrome, can occur until 5–6 years of age.

### 3.1. Pathogenesis

The increased risk of seizures in children and young adults with DS has been linked to several mechanisms, including frontal and temporal lobe hypoplasia, dyskinesia of dendrites, abnormal neuronal lamination, decreased neuronal density, especially of inhibitory interneurons with a GABAergic inhibitory function, alteration in cell membrane ion channels, specifically of the glutamatergic receptor GluR5 (encoded in chromosome 21), other metabolic disorders associated with the condition of trisomy 21, and to complications derived from hypoxia and/or vascular damage due to a higher prevalence of congenital heart disease and the perioperative risks of its correction [32,37,38,39,40,41,42,43].

### 3.2. Febrile Seizures

Febrile seizures are defined as symptomatic seizures occurring in the context of fever. As in the general population, they are the most frequent neurological phenomenon in childhood between 6 months and 5–6 years. However, the prevalence of febrile seizures in children with DS seems to be lower than in the general population (2.5% vs. 7–9%) [38,44]. The reasons for this partial protection in children with DS are currently unknown. In the general population the risk of developing epilepsy after a febrile seizure is estimated to be 2 to 3 times higher than in those without [45]. The prognosis after febrile seizures in DS has not been established. It is also unknown if a first-degree relative with history of febrile seizures, one of the main risk factors for the development of febrile seizures in the general population [38,45,46], is also a risk factor in children with DS.

In contrast, untriggered seizures meeting diagnostic criteria for epilepsy are more frequent in children and young adults with DS. Generalized epilepsies are more frequent than focal epilepsies in the first two decades of life [37].

### 3.3. Other Epileptic Seizures

-Infantile spasms and West syndrome: In early childhood, infantile spasms (flexor, extensor or flexo-extensor spasms) affecting the neck, trunk and/or proximal regions of the extremities occur in the first 2 years of life with a peak incidence at 5 months. They usually last less than 5 s and appear in series of episodes that can last from less than 1 min to more than 10 min. They have a morning predominance. Children with DS have muscle hypotonia, which complicates the diagnosis of the spasms, specially in the case of subtle seizures [36].

West syndrome is characterized by infantile spasms in the context of hypsarrhythmia or a modified hypsarrhythmia pattern in the electroencephalogram. It associated with a risk of neurodevelopmental regression, and is the most frequent type of epilepsy (it represents between 6 and 46% of all epilepsy cases) in children with DS, with an estimated prevalence ranging from 0.6 to 13% [37,39,46,47,48] (Table 2). Fortunately, and despite its higher prevalence in DS, the response to early pharmacological treatment with adrenocorticotropic hormone (ACTH) seems to be better than in the general population, with less risk of seizure recurrence in the long term. Complete remission is achieved in more than 90% of cases. As in the general population, a EEG flare-suppression pattern and a delay in diagnosis are risk factors for the development of an epileptic encephalopathy [37,39,47,49,50,51].

-Generalized tonic-clonic seizures (GTCS): The second most frequent type of epileptic seizures in children with DS are GTCS. They are more frequent than generalized non-motor seizures and have more severe complications due to a higher risk of falls and bronchoaspiration. GTCS are easier to diagnose in the anamnesis.

Other types of epileptic seizures with a motor component frequently described in the context of other causes of intellectual disabilities such as reflex epileptic seizures to sensory stimuli, mainly auditory, are infrequent in context of DS, occurring in approximately 1% in this age range [40,52,53,54].

-Generalized non-motor seizures, typical, atypical and myoclonic absences have also been described in children with DS [35,55]. It is not possible to estimate the frequency of these seizures as most of the published studies on this subject are isolated cases or series of cases with small sample sizes. Under-diagnosis is to be expected given the complexity of their correct identification by semiology in the DS population.-Lennox-Gastaut syndrome is an epileptic syndrome characterized by the triad of drug-resistant epilepsy with multiple seizure types, typically atonic seizures and atypical absences during wakefulness and mainly nocturnal tonic seizures. The EEG typically shows diffuse slow spike-wave or polyspike-wave discharges during wakefulness and bursts of diffuse fast rhythms at 10–20 Hz during sleep. In the general population, the incidence of Lennoux-Gastaut syndrome is 0.1–0.28/100.000, but there is a lack of reliable data in children with DS to estimate an incidence. Of note, the probability of progression from West syndrome to Lennoux Gastaut syndrome seems to be lower in children with DS [36]. Available evidence from small case series point to a later onset and a higher frequency of reflex seizures compared to the general population in DS-associated Lennox Gastaut syndrome [35,56].

Epilepsy in children with DS can hinder cognitive and motor development and significantly impact their quality of life, especially if the diagnosis is delayed or the therapeutic management is suboptimal [49,57]. An early and accurate diagnosis and treatment could result in improvements in cognitive performance. Specifically, an improvement in verbal communication skills has been described in children with DS and West Syndrome after early treatment with ACTH combined or not with antiepileptic drugs such as topiramate or vigabatrin [58].

### 3.4. Electrophysiological Characterization

Nonspecific abnormalities such as mild slowing of background activity in children with DS have been described more frequently than in the general population [59]. Consequently, quantitative electroencephalography analysis can reveal differences between children with DS and euploid children in the alpha, delta and beta rhythms [60].

The critical and intercritical EEG patterns vary significantly depending on the type of seizure, and the existence or not of a specific epileptic syndrome, which often are associated with a characteristic EEG pattern. In typical absences, paroxysmal epileptiform activity is usually detected in the form of generalized paroxysm of spike-wave with a frequency between 2.5 and 4 Hz; frequencies below 2.5 Hz in atypical absences; and polyspike-wave paroxysms with frequencies between 3 and 6 Hz in myoclonic absences [55]. West syndrome, by definition, is associated with the presence of hypsarrhythmia, defined by the presence of continuous discharges of slow waves, spikes or sharp waves, without synchronization between both hemispheres and of high voltage, giving the sensation of an absolute disorder of the electroencephalogram [39,50,51,61].

### 3.5. Response to Antiepileptic Drugs

There are no clinical trials to support the use of one antiepileptic drug over another in the DS population. Therefore, in the absence of specific guidelines, the same assumptions and treatment options are usually made as in the general population. Newer antiepileptic drugs have not proven to be more effective but are generally better tolerated than the classical or first generations drugs.

Due to the frequent cardiac malformations in subjects with DS, it seems reasonable to avoid antiepileptic drugs with a higher proarrhythmogenic potential (although data to support this is lacking). If the epilepsy cannot be precisely classified as focal or generalized, broad-spectrum drugs should be used [62]. In cases of generalized epilepsy with or without myoclonic seizures, valproic acid remains an alternative, with analytical controls and evaluation of the risk of inducing cognitive and motor side effects. Lamotrigine in the absence of myoclonic seizures is potentially a good alternative due to its broad-spectrum profile. In this case, the need for analytical controls and the possibility of pharmacological interaction should be considered. In the case of West syndrome, treatment with ACTH combined or not with antiepileptics drugs such as vigabatrin, valproic acid, topiramate or levetiracetam should be used [36,58].

## 4. Epilepsy in Children over 5 Years and Young Adulthood

### 4.1. Pathogenesis

Epilepsy in DS has a clear bimodal distribution. Reliable information about epilepsy with onset in later childhood, young adulthood and middle ages is lacking. First, there is no sufficient information about the evolution of the aforementioned seizures and epileptic syndromes in early childhood. Second, the prevalence estimates of epilepsy in the adolescents and young adults with DS reported in the literature are variable, ranging from 8% to 14% [63]. The reasons for this variability might include: (1) different origin of the data (routine medical care vs epidemiological registry or health plans); (2) difficulties in the anamnesis and differential diagnosis (to rule out other causes); (3) risk of overinterpretation of variants of normality and/or non-epileptiform abnormalities in EEG as epileptiform or vice versa when interpreted outside an adequate clinical context; and (4) most importantly, the age ranges included.

Finally, it is not established whether there is a higher risk of new onset epilepsy in DS with respect to the general population in this age range. The most frequent etiologies of epilepsy in this age range are comparable to the general population, e.g., structural damage such as trauma or ischemic (among others).

### 4.2. Clinical Characterization

There is a large variability in the type of untriggered seizures in older children and adults with DS between 5 and 40–45 years of age. These epileptic seizures might be focal, with and without altered level of consciousness or generalized motor and non-motor seizures [1,16,20]. Seizure semiology varies depending on the age of onset and etiology as in general population. Focal or generalized nonmotor seizures will probably be underdiagnosed in this age group compared to early childhood and older age and cognitive impairment, because in this age range there is less continued external supervision by caregivers.

### 4.3. Electrophysiological Characteristics

The use of routine video-EEG is indicated for the differential diagnosis of episodes that could potentially suggest epileptic etiology. The diagnostic sensitivity of routine video-EEG is generally low. It increases the closer it is performed to the seizure, and with longer recording durations, ideally including sleep recording. In this population worse collaboration is frequent on activation maneuvers, such as hyperventilation and photic stimulation.

EEG characteristics will vary according to the type of epilepsy and are also highly variable. There are frequent anomalies with no clear epileptiform significance such as generalized and/or focal slowing that require cautious interpretation. In short, EEG findings must always be interpreted after careful consideration of the clinical context.

### 4.4. Response to Antiepileptic Drugs

The choice of broad-spectrum drugs is recommended if the focal and/or generalized onset of seizures cannot be clearly established. If a focal onset is clear, drugs with exclusive indication for focal epilepsy may be used as in the general population. Among the broad-spectrum drugs, the newer drugs have a better safety profile with comparable efficacy and should therefore be recommended. The choice should also take into account the existing comorbidities, the potential need for blood level monitoring and the potential difficulties in adherence to a given treatment. Levetiracetam (LEV) is a priori a good candidate since it is effective for both focal and generalized epilepsies including myoclonic seizures. It is widely available both orally and intravenously and has a very low risk of drug interaction. Tolerance of LEV in the DS population as in other cases of intellectual disability is similar to the general population, and the same precautions should be applied: avoid in the presence of moderate-severe behavioral symptomatology and/or renal insufficiency.

## 5. Epileptic Seizures in Late Adulthood (≥40 Years) and Their Relationship with AD

As mentioned before, the second peak in epilepsy incidence is closely associated with the development of symptomatic AD.

### 5.1. Pathogenesis

Several factors might account for this increased risk. The triplication of the amyloid precursor protein gene, which is encoded in chromosome 21, leads to amyloid overproduction. The extracellular accumulation of amyloid-β (Aβ) plaques induces synaptic degeneration, circuit remodeling and abnormal synchronization of neuronal networks (Aβ_1-42_ alters the membrane potential and induces ionic channel dysfunctions affecting selectively to potassium currents). Thus potentially favoring increased cortical irritability and inducing a pro-epileptic effect [2,9,29] in individuals who, in addition, have neurodevelopmental structural and cell membrane function abnormalities from birth. The dysfunction of cholinergic and glutamatergic systems (impairment of glial and neuronal uptake of glutamate) in the context of AD impact neuronal excitability and epileptic pathophysiology [12]. Interestingly, soluble Aβ has been associated with decreased GABAergic inhibition and increased hyperexcitability. Tau protein has been considered a necessary mediator of the epileptogenic effects of Aβ, increasing significantly extracellular glutamate and inducing dysfunction of the NMDA (N-metil-D-aspartate) receptors. In turn, Aβ-driven overexcitation, mainly in the entorhinal-hippocampal circuits would lead to compensatory inhibitory changes in hippocampal networks, restricting synaptic plasticity and contribute to learning and memory deficits and accelerating the cognitive impairment in AD [64]. Early locus coeruleus degeneration in DSAD, could be particularly relevant since norepinephrine (the locus coeruleus is the main noradrenergic nucleus of central nervous system) has potent anticonvulsant effects [65]. In addition, the vascular burden associated with amyloid angiopathy, which is more frequent and severe in DSAD than in ADAD [66], could also partly explain the increased risk of developing epileptic seizures [67].

The relationship between Aβ and epileptogenesis is bidirectional. Both the Aβ oligomeric forms and plaques have direct proepileptic effects. In turn, the presence of epileptic seizures and intercritical epileptiform activity favors Aβ deposition [68,69,70]. In the general population, reduced CSF Aβ1-42 levels are more frequently found in individuals with late-onset epilepsy of unknown etiology (LOEU) than in those without epilepsy and have a higher risk of progression to AD dementia 3 to 8 years after seizure onset [71]. Another interesting finding of unknown significance is that of higher mid-life Aβ deposition measured by PET in individuals with childhood-onset epilepsy [72].

### 5.2. Clinical Characterization

The prevalence of untriggered seizures in DS increases dramatically with age after age 45, in parallel with the emergence of symptomatic AD. The prevalence of epilepsy can be as high as 46% in those over 50 years of age, driven by the patients with symptomatic AD. This association is so close that the occurrence of a first episode of an untriggered seizure after the age 40–45 years is highly suggestive of symptomatic AD [16,17,21,29,63,73,74,75].

AD is also an independent risk factor for the development of epilepsy in the general population (prevalence estimates ranging from 10 to 22%) [76,77], and even more so in autosomal dominant AD (prevalence ranging from 19.1 to 47.7%) [67,78]. However, epilepsy is significantly higher in DS, with prevalence estimates ranging from 46 to 84% [16,73]. Thus, the risk is at least 10 times higher in DS than in sporadic AD, and significantly higher than in autosomal dominant AD, including AD patients harboring APP duplications [11,12,17,29,32,67,74,79] (Table 3).

It is important to note that a single untriggered epileptic seizure in the context of AD in DS fulfills the criteria of epilepsy because the risk of recurrence after seizure can be estimated in up to 70% in the next 7.5 months [81]. Antiepileptic treatment should be thus promptly initiated and maintained (Table 1). 

The most frequent epileptic seizures in the context DSAD are GTCS and myoclonic seizures (Table 3), which usually coexist. At onset, the seizures have a similar time pattern as in juvenile myoclonic epilepsy, with clear morning predominance upon awakening. The frequency of focal seizures with altered level of consciousness without a motor component, the most frequent type of epileptic seizure in sporadic AD, is low in DS [29]. On the contrary, myoclonic seizures, which usually affect the trunk and the upper extremities symmetrically are relatively common [82]. They often present in a series of seizures, but the frequency of massive and/or fall-inducing myoclonus is low [11,20,24,32,73,83,84,85,86].

Due to these specific clinical characteristics of this epilepsy associated with symptomatic AD in DS, a specific term has been coined, “late-onset myoclonic epilepsy in DS” (LOMEDS) [11,12,17,24,31,82,87,88] (Figure 2). The clinical course of this myoclonic epilepsy seems to be similar to other progressive myoclonic epilepsies, in which myoclonus becomes more frequent, more resistant to treatment and dissociates from the EEG as the cognitive decline progresses [12,29]. The development of multifocal myoclonus, sometimes induced by movements or less frequently by sensory stimulus increases the risk for falls. Myoclonus is frequently refractory to treatment and does not have EEG correlates, suggesting a non-cortical origin of these myoclonic jerks. Tremor and disabling cerebellar ataxia, which also coexist with DSAD, especially in the context of LOMEDS, also contributes to the risk of falls [53].

Symptomatic AD usually precedes the onset of the first epileptic seizure, usually by 2–3 years. However, in up to 5% of cases, it can be the first clinical manifestation of symptomatic AD [11,12,16,40,74,83,89]. We note that myoclonic seizures are probably underdiagnosed, which might affect rigorous evaluation of the temporal evolution of seizure onset with respect to that of cognitive impairment due to AD. More advanced AD dementia is associated with a greater risk of developing seizures. In sporadic AD and ADAD, an earlier age of onset is associated with an increased risk for the development of seizures, but this remains to be proven in DSAD [2,29,63,90].

The development of epileptic seizures in DSAD has been associated with a more rapid cognitive and functional decline [12]. Epilepsy is also considered as a risk factor for more frequent hospital admissions. Finally, epilepsy is also an independent risk factor for mortality in adults with DS [6,10,57,89,90,91,92,93,94].

### 5.3. Electrophysiological Characteristics

Diffuse slowing of background activity is the most frequent finding on surface EEG in adults with DS, both in persons with and without epilepsy [9,16,17,95,96,97]. This slowing of background activity is more evident in prodromal AD and even more in the dementia stage [20,29].

There is a poor correlation between the severity and control of epilepsy and surface EEG findings [17]. As in the general population, the diagnostic performance of the routine video-EEG for the detection of intercritical epileptiform abnormalities is low but increases if it is performed close to occurrence of the seizure, and if performed soon after awakening. A night sleep recording might also increase the sensitivity. A positive photo-paroxysmal response could be an indicator of a worse pharmacological control of myoclonic seizures [11,24,32]. Among the epileptiform anomalies, the best characterized and most frequent are the presence of generalized intercritical or critical paroxysms of spike-wave or polyspike-wave paroxysms (Figure 3), which may or may not be related to myoclonus [2,16,17,24,97]. The presence of focal intercritical paroxysms with frontal predominance is less frequent and even more that of temporal paroxysms [17,88]. As in the general population, epileptiform abnormalities of uncertain significance can be detected in the absence of evidence of seizures and in the context of symptomatic AD (described in 28% of sporadic AD cases) [77,98]. The frequency of such epileptiform activity without associated clinical seizures and the potential cognitive impact is not yet elucidated neither in sporadic AD nor in DSAD.

The association between EEG findings and myoclonus supports that these are caused by generalized motor epileptic seizures [87]. This might have therapeutic implications, as myoclonus with a clinical or electroclinical pattern similar to that of juvenile myoclonic epilepsy (JME) might prompt antiepileptic treatment. However, more data is needed to support this guidance.

The diagnostic performance of EEG increases with longer record length and with sleep recording, being frequent to detect intercritical and critical epileptiform abnormalities in nocturnal EEG recording (Figure 4).

### 5.4. Response to Antiepileptic Drugs

The different electroclinical characteristics of epilepsy in sporadic AD and DSAD might warrant different treatments. In sporadic AD, the lower frequency of myoclonus and higher frequency of focal epileptiform activity has led to the use of a greater variety of drugs without considering a potential worsening of myoclonus (e.g., when using lamotrigine) or worse seizure control (when using drugs with exclusive indication for focal epilepsy). However, in DSAD, the use of broad-spectrum antiepileptic drugs with antimyoclonic effect is recommended. Monotherapy at the lowest possible dose with a very slow dose scalation is preferred. Good seizure control of both GTCS and myoclonic seizures can be achieved in more than 80% of LOMED cases [2,11,12,19]. The drug of first choice is currently levetiracetam (LEV), which has proven its usefulness in different generalized myoclonic epilepsies [2,29,31,83] and has few drug interactions. Tolerance to LEV seems to be comparable in the DS population with respect to the general population, being infrequent the need for its withdrawal at low-intermediate doses due to behavioral problems. The low interaction profile is important as DSAD patients are very often on polytherapy. Valproic acid is an alternative for generalized myoclonic epilepsy in DS. It also achieves good control of GTCS and myoclonic seizures [29,32]. If good seizure control is not achieved with first line treatments (LEV or VPA), the best alternative is to combine them at the lowest possible dose (Table 4). Among the newer antiepileptic drugs, brivaracetam, with a mechanism of action similar to LEV, but with a better profile of behavioral side effects, might be a good therapeutic alternative in the few cases of intolerance to LEV. However, it still lacks an approved indication for generalized epilepsies at the present time, and for the time being, should be used as an add-on therapy with VPA. Perampanel, a selective non-competitive AMPA receptor antagonist, with an approved indication for use in GTCS and with encouraging results in myoclonic seizure control, seems to be a good alternative as an add-on therapy to LEV or VPA when bitherapy is required. The tolerance in DS is comparable to that in the general population. On the other hand, it is very important to avoid the use of sodium channel blocking antiepileptic drugs, which have been shown to aggravate myoclonic seizures [12,99].

In summary, the use of broad-spectrum antiepileptic drugs without the potential to worsen myoclonus is recommended, with newer antiepileptic drugs being preferred to the classic ones due to their better adverse effect profile (Table 4). As noted, the response to carefully selected antiepileptic drugs is satisfactory at low doses in most cases for both GTC and myoclonic seizures. This behavior is similar to that described in epilepsy associated with sporadic AD where good seizure control is relatively easy to obtain with low dose of antiepileptic drugs. The appearance of myoclonus refractory to the initially established and effective antiepileptic treatment should raise the possibility of the evolution to a progressive myoclonic epilepsy where not all myoclonuses are of epileptic origin. In the latter case, it is important to perform a detailed anamnesis of the myoclonus including a calendar of episodes, obtain home recorded videos of the episodes and perform prolonged recordings with video-EEG.

A personal history of epilepsy, whether LOMEDS or not, does not contraindicate the initiation of acetylcholinesterase inhibitor treatment, nor does it oblige to its suspension (drugs widely used in DSAD) [100]. Instead, it is recommended to be more restrictive due to lack of supporting evidence in DSAD and more cautious with the use of memantine, N-methyl-d-aspartate (NMDA)-receptor antagonist, in case of LOMEDS. It is also important to identify and specifically treat comorbidities such as sleep disorders, e.g., OSA, which can worsen seizure control in DSAD (Table 4).

## 6. Recommendations in Clinical Practice and Future Directions

The first step should always be a detailed semiology of the episode suggestive of epileptic seizure. Of note the interview of the patient is usually less informative than in the general population (due to the intellectual disability and possible to the type of seizures associated with the syndrome). Home videos and a routine video-electroencephalogram (EEG) are always recommended after a first episode (as soon as possible to increase its diagnostic performance). Screening for possible triggers by performing a blood test including at least a hemogram, blood glucose, renal function and ionogram; urine sediment should always be performed as well as a cranial tomography or magnetic resonance imaging. Early identification and treatment of seizures is essential. After the onset of antiepileptic treatment, monitoring of its response, together with the correct identification and treatment of medical comorbidities, especially sleep problems, that may worsen seizure control is essential (Table 4).

There are several important gaps in knowledge in LOMEDs with potential important implications. There is a need to correctly identify the risk factors for epilepsy in this particular population: 1- there is a lack of information to know whether the risk of epilepsy both in childhood and in relation to the development of symptomatic AD varies depending on whether it is a subject with complete trisomy versus a mosaic or translocation; and 2- the role of the main genetic risk factor for sporadic AD which is the ApoE4 allele, linked to earlier onset AD and increased risk of epilepsy in sporadic AD, in the risk of developing LOMEDS is not known. From a diagnostic perspective, the temporal relationship of epileptic seizures and symptomatic AD in DSAD should be better established. Optimizing the use of video-EEG and, potentially, performing sleep recordings (which would enable the screening of OSA) could be useful and of significant clinical relevance. It is also expected that in the future the development of new technologies (e.g., mobile apps, smartwatches...) may play a role in the objective monitoring of epilepsy and thus optimize diagnosis and treatment. The use of individual devices such as those wearables approved by the FDA and EMA for detection of seizures could potentially be useful for remote monitoring of patients and alerting to evidence of new seizures. However, there is a lack of evidence to recommend their use in clinical practice in the DS population and the detection capacity for seizures other than generalized tonic-clonic seizures still needs to be optimized. The same applies to home EEG seizure-detection systems that need to be validated.

More importantly, epilepsy drug treatment recommendations for LOMEDS are based on clinical practice experience rather than clinical trials. Clinical trials should establish if treatment of myoclonus delays or prevents GCTS in DSAD. These trials should also evaluate the potential benefit that antiepileptic drugs on AD progression. Furthermore, given the feed forward loop between AD pathophysiology and hyperexcitability, antiepileptic treatment could be tested in secondary prevention trials in DS. Of note similar trials are underway in the general population (An Investigation of Levetiracetam in Alzheimer’s Disease (ILiAD), Levetiracetam for Alzheimer’s Disease-Associated Network Hyperexcitability (LEV-AD), Treating Hyperexcitability in AD With Levetiracetam and Network-Level Mechanisms for Preclinical Alzheimer’s Disease Development. ClinicalTrials.gov accessed on 21 June 2021). Finally, in the general population, OSA treatment improves seizure control in LOEU. Similar studies should be performed in LOMEDS, especially given the very high prevalence of OSA in DS [99], which is even higher in symptomatic AD [100].

## 7. Conclusions

Epilepsy is one of the most frequent comorbidities in DS and has a clear bimodal distribution. The first peak occurs in early childhood in relation with the neurodevelopmental abnormalities associated with the syndrome and the second in the sixth decade in relation with symptomatic AD. Therefore, the clinical characteristics of epilepsy vary widely throughout the lifespan. Epilepsy has important clinical and functional consequences, and it is an independent risk factor for both hospitalization and mortality. LOMEDS, the epilepsy associated with symptomatic DSAD, has specific clinical and electrophysiological features, which differ from the epilepsy associated with sporadic AD, and warrants specific therapeutic management. LOMEDS is probably underdiagnosed and not adequately treated in many cases. Future studies are needed to confirm whether an adequate therapeutic treatment might improve the functional and vital prognosis in LOMEDS.

## Figures and Tables

**Figure 1 jcm-10-02776-f001:**
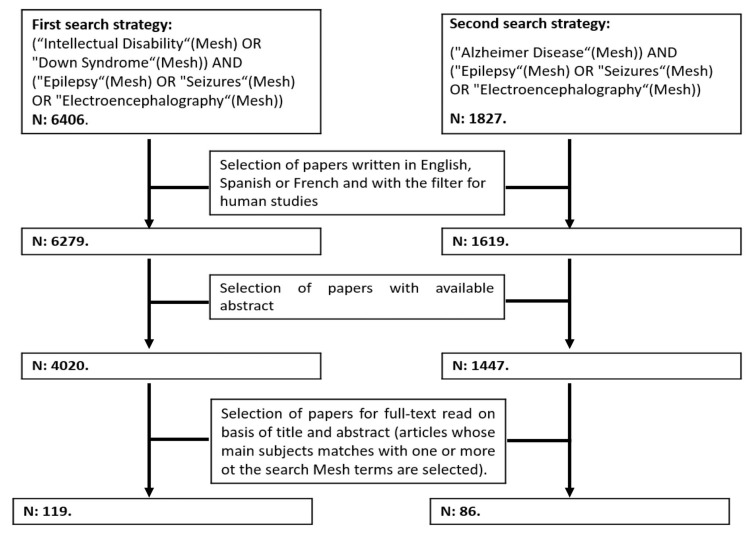
Summary of the search strategies used in PubMed and WOS and information on the selection of articles for full reading.

**Figure 2 jcm-10-02776-f002:**
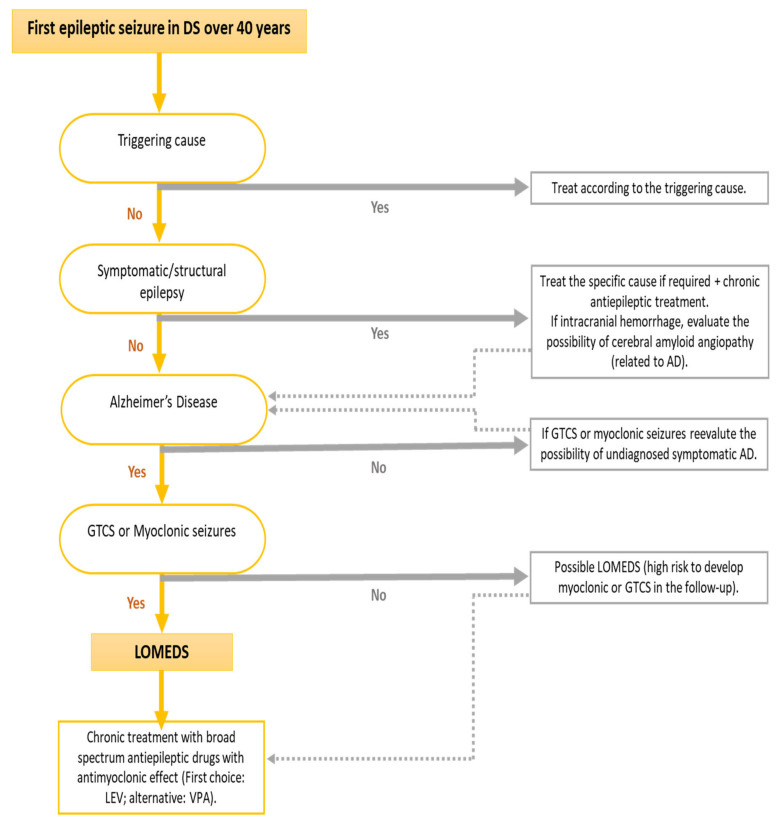
Flowchart including the main diagnostic recommendations for late onset seizures in Down syndrome. Recommendations based on expert opinion. (AD: Alzheimer’s Disease. GTCS: Generalized tonic-clonic seizures. LOMEDS: Late Onset Myoclonic Epilepsy in Down syndrome. LEV: levetiracetam. VPA: valproic acid).

**Figure 3 jcm-10-02776-f003:**
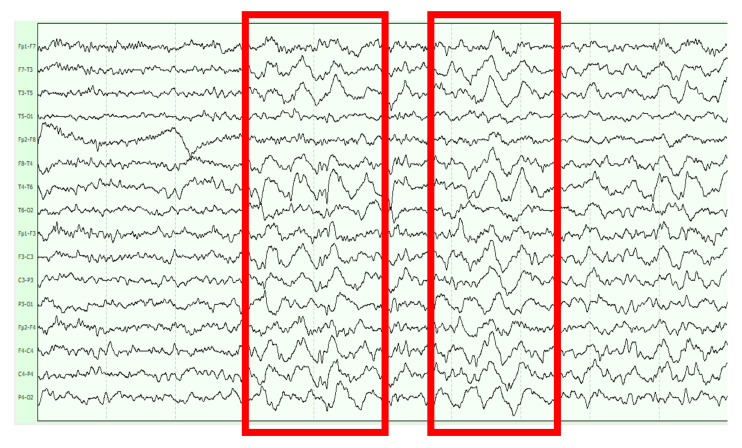
Paroxysms of generalized spike-wave and polyspike-wave epileptiform activity in bipolar longitudinal montage of EEG.

**Figure 4 jcm-10-02776-f004:**
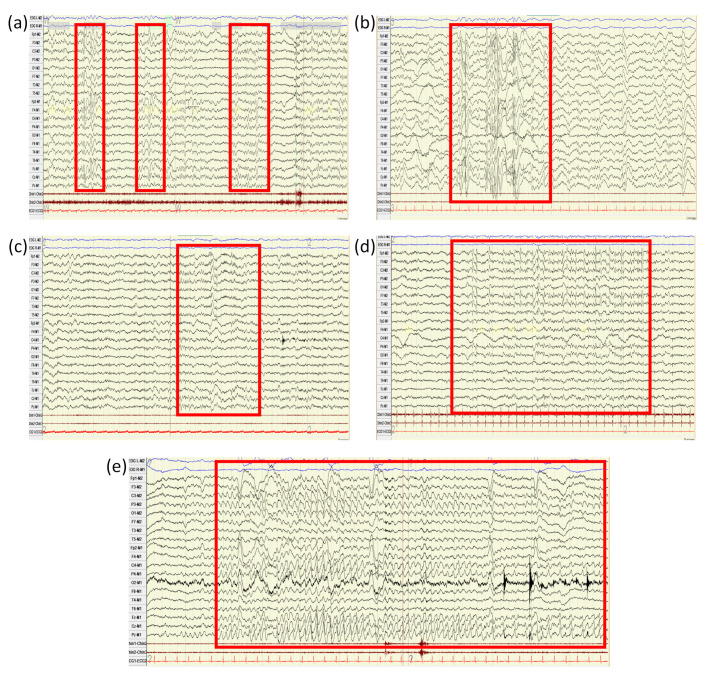
Prolonged electroencephalogram recordings including nocturnal sleep in patients with DSAD: (**a**) polymorphic generalized triphasic waves with maximum frontal amplitude in N1 phase of sleep; (**b**) generalized polyspike-slow waves in N3 phase of sleep; (**c**) paroxysmal activity in the form of sharp waves with maximum amplitude in the left frontal region; (**d**) spikes at 3 Hz frequency in left fronto-temporal region during N2 phase of sleep and (**e**) generalized critical epileptiform activity upon awakening.

**Table 1 jcm-10-02776-t001:** Definitions of seizures (triggered and untriggered) and epilepsy according to the International Epilepsy League.

**Seizure:**
Transient occurrence of signs and/or symptoms caused by excessive or simultaneous abnormal neuronal activity in the brain.
Untriggered/Unprovoked seizures:
Those occurring in the absence of a specific clinical context that reduces the individual’s seizure threshold.
Triggered/Provoked seizures:
Those occurring in a specific clinical context that reduces the individual’s seizure threshold (e.g., infection or fever, acute brain injury whether ischemic/hemorrhagic/inflammatory-infectious or traumatic, consumption of toxic substances, metabolic disturbances, abrupt withdrawal of medications that may promote seizures such as benzodiazepines, sleep deprivation…).
**Epilepsy:**
Refers to one of the following conditions:
(1) occurrence of at least 2 unprovoked seizures separated by ≥24 h;
(2) occurrence of an unprovoked seizure and an estimated probability of 10-year seizure recurrence similar to the overall recurrence after 2 unprovoked seizures (estimated >60%) or
(3) diagnosis of a specific epileptic syndrome.

**Table 2 jcm-10-02776-t002:** Summary of the main characteristics of epileptic seizures according to age of onset in DS. (GTCS: generalized tonic-clonic seizures and AD: Alzheimer’s Disease).

**Epilepsy in early childhood:**
Peak incidence in the first 2 years. Febrile seizures are the most common seizures (but less than in typically developing children). Infantile spams/West syndrome are the most frequent untriggered seizures/epilepsy with good response to treatment. GTCS are frequent in the first 2 decades, but other seizure types are more infrequent.
**Epilepsy in late adulthood:**
Peak incidence after the development of AD dementia. Functional repercussion and potential faster cognitive deterioration. Generalized myoclonic epilepsy with GTCS and myoclonic seizures. Good initial response, but high risk of progression to refractory progressive myoclonic epilepsy.

**Table 3 jcm-10-02776-t003:** Clinical and electrophysiological differences between epileptic seizures in relation to sporadic AD vs autosomal dominant Alzheimer’s disease (ADAD) and DSAD.

	Sporadic AD	ADAD	DSAD
**Prevalence**	1.5–12.7% [80]	2.8–47% [67,81]	41.1–75% [29,74]
Most frequent type of seizures	Focal seizures with altered level of consciousness and without motor symptoms	82% generalized motor seizures, 8% focal with generalization, 8% focal with impairment of consciousness and 2% focal [67]	Myoclonic and GTCS
EEG most frequent epileptiform findings	Focal temporal slow and acute waves or spikes	Spike waves, spikes, rapid slow waves focal or generalized [67]	Generalized polyspike-waves
Temporal relation between AD onset and epilepsy onset	First seizure in dementia state for AD	Higher risk in carriers without cognitive impairment but majority first seizure in dementia state for AD [78,81]	First seizure in dementia state for AD
Clinical progression	Good seizure control after onset of antiepileptic drug	Good seizure control after onset of antiepileptic drug. In last stages of the disease possible progression to refractory myoclonus	Initial good response and frequently progression to refractory myoclonus

**Table 4 jcm-10-02776-t004:** General and specific recommendations on diagnosis, treatment and follow-up of subjects with epilepsy, LOMEDS or not. Recomendations based on clinical practice experience. (AED: antiepileptic drugs; LEV: levetiracetam; VPA: valproic acid; OSA: obstructive sleep apnea).

Epilepsy in DS: General Recommendations	Specific Recommendations in LOMEDS
❑ Structured anamnesis with patient and caregivers with neurological exam. ❑ Check for potential triggers: Metabolic disturbances, infections, sleep deprivation, medication failure, brain traumatic injury, e.g., ❑ Video-EEG as soon as possible after episode or relevant changes in seizure semiology. ❑ Request home videos of episodes. ❑ Chronic treatment with AED based on risk of seizure recurrence. ❑ Unless clearly established focal origin of seizures use broad-spectrum AED. ❑ Monotherapy is preferred. ❑ Monitoring seizure frequency, treatment compliance, tolerance and possible interactions. ○ Blood test monitoring if required.	❑ Myoclonus not always have an epileptic etiology (e.g., metabolic origin). ○ Attention to both underdiagnosis and overdiagnosis. ❑ Use broad spectrum drugs (first choice: LEV, alternative: VPA). ○ Monotherapy is preferred. ○ Start low and go slow. ○ Avoid sodium channel blockers and gabaergic antiepileptic drugs (worsening of myoclonic seizures). ❑ Epilepsy is not a criterion either for withdrawing or not initiating treatment with anticholinesterase drugs. ❑ Evaluate the coexistence of associated sleep pathology (mainly OSA), which can aggravate seizure control.

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
