# Peer review of "Epilepsy in Down Syndrome: A Highly Prevalent Comorbidity"

_jcm, 2021, doi:10.3390/jcm10132776_

Round 1

Reviewer 1 Report

This study conducted a narrative review of the literature to summarize the current knowledge about the clinical and electrophysiological characteristics, diagnosis and treatment of epileptic seizures among patients with DS and provided insightful recommendations for clinical practice and future directions. I have a few minor questions/comments, noted below. 

  1. In figure 1, it would be beneficial if the authors could add reasons for exclusion of papers at the full-text screening stage. Using the PICOS criteria for reasons for exclusion may an option to explore.
  2. I was just curious if a quality appraisal was conducted for the papers included in the review. 

Author Response

This study conducted a narrative review of the literature to summarize the current knowledge about the clinical and electrophysiological characteristics, diagnosis and treatment of epileptic seizures among patients with DS and provided insightful recommendations for clinical practice and future directions. I have a few minor questions/comments, noted below. 

  1. In figure 1, it would be beneficial if the authors could add reasons for exclusion of papers at the full-text screening stage. Using the PICOS criteria for reasons for exclusion may an option to explore.

First of all, thank you for your suggestions.

I will respond to your comments: 

  1. When searching and selecting articles, we did not strictly follow the PICO criteria. For any future work we will try to apply them as we are aware that they increase the robustness of the search and selection of papers.

The selection of articles for full reading was after reading the abstract and verifying that the main subject matter corresponded to the Mesh terms that we had established for the search strategy. If the papers made reference to but did not have any of the search criteria as a central topic, the article was not read in its entirety. Nor did we proceed to the complete reading of those articles that appeared in the initial search corresponding to animal models despite using the filter of human-only studies.

Following your recommendations, in the figure itself we have added a sentence that aims to clarify our way of selecting articles and that completes the text.

2. I was just curious if a quality appraisal was conducted for the papers included in the review. 

  1. Despite having a considerable number of articles on the subject under study, we did not consider using quality criteria for an initial filtering of the papers. Most of the articles that study epilepsy in the adult population with Down syndrome correspond to case reports or case series and there are few robust studies in this area. As the main focus of our review was precisely epilepsy in adults with Down syndrome, we decided to proceed to a critical reading of all the papers that could potentially address the subject without a quality filter. If the subject under study had been different, such as epilepsy in sporadic forms of Alzheimer's disease, we would have applied these quality criteria in the selection, since the robustness of the papers on this subject is much higher than that of epilepsy associated with Alzheimer's disease in Down syndrome.

Reviewer 2 Report

The authors seek to raised awareness and provide a better understanding of epilepsy in DS from families, caregivers, and clinicians to enable earlier  diagnoses and better treatments for individuals with DS.

Writing Quality: The manuscript is well written and free of grammatical mistakes.

Reviewed concepts:

Type of seizures associated at different ages (bimodal) with DS. Well done and informative.

At the bimodal prevalence times in life there was good discussion of the Pathogenesis, Clincial types of seizures, electrophysiological observtions in the clinic  and epileptiform response to drugs. There are already a lot of revies in the literature on DS and Epilepsy so I feel the need to make reccomendations o to help this review better stick out.

Please provide more discussion on DS patient outcomes to treatments. This would be very beneficial to the readers especially if the intent is to raised awareness in families, caregivers, and clinicians. This may require some interviews with physicians and could be empowered by the LoDowns consortium or by the NIH DS-Connect portal. Alternatively, discussing the general population might have to serve as proxy for what to expect in DS children or adults.

Figure 2 was very nice, and more EEG patterns would be briliant if the authors could provide spike  waves, spikes,rapid slow waves focal or generalized . Also please consider  discussion on whether if these patterns are diferrent for individuals with DS vs non-DS + Epilepsy at the two bimodal prevelance times.

Figure 3 might need font enlargements. Please check for readability in potential final drafts.

In section 4, the recommendations are very good as it is extremely hard to diagnose epileptiform activity in general. Often EEGs may miss the episode. The authors might consider discussion on novel  wearable EEG microdevices and translation to a family with a child with DS.

The authors could also discuss other (home friendly) technologies that might help diagnose epileptiform activity without having to make a direct observation at a clinic clinic.

Overall the organization is just right and the topic was well reviewed.

Author Response

The authors seek to raised awareness and provide a better understanding of epilepsy in DS from families, caregivers, and clinicians to enable earlier  diagnoses and better treatments for individuals with DS.

Writing Quality: The manuscript is well written and free of grammatical mistakes.

Reviewed concepts:

Type of seizures associated at different ages (bimodal) with DS. Well done and informative.

Thank you very much for your suggestions.

At the bimodal prevalence times in life there was good discussion of the Pathogenesis, Clincial types of seizures, electrophysiological observtions in the clinic  and epileptiform response to drugs. There are already a lot of revies in the literature on DS and Epilepsy so I feel the need to make reccomendations o to help this review better stick out.

Please provide more discussion on DS patient outcomes to treatments. This would be very beneficial to the readers especially if the intent is to raised awareness in families, caregivers, and clinicians. This may require some interviews with physicians and could be empowered by the LoDowns consortium or by the NIH DS-Connect portal. Alternatively, discussing the general population might have to serve as proxy for what to expect in DS children or adults.

We have extended the section on pharmacological treatment management in both children and adults following your suggestions. These recommendations have been based on our clinical practice, since we are neurologists who regularly treat adults with Down syndrome as part of a health plan.

Figure 2 was very nice, and more EEG patterns would be briliant if the authors could provide spike  waves, spikes,rapid slow waves focal or generalized . Also please consider  discussion on whether if these patterns are diferrent for individuals with DS vs non-DS + Epilepsy at the two bimodal prevelance times.

On the other hand, we have incorporated more examples of critical and incritical abnormalities (new Figure 3) that we have obtained by means of nocturnal EEG recordings. The best recordings have been obtained in this context. We believe that the images reflect well the different anomalies detailed at the bottom of the figure.

In Table 1 we compare the most frequent EEG findings in different forms of AD so we have not incorporated more information regarding this in the text. On the pediatric part, given the greater variability of EEG patterns, we believe that a summary description as currently available may be sufficient. We do not have EEG images from our clinical practice in childhood. At the same time, we believe that the published literature on childhood epilepsy in DS is more robust than epilepsy associated with DSAD and therefore, we wanted to focus more on the latter.

Figure 3 might need font enlargements. Please check for readability in potential final drafts.

We have increased the size of Figure 4 (previously Figure 3) as recommended. When printed at the current size, the figure maintains good resolution. If any adjustments are required in the final edition we will make sure to keep the image with good quality.

In section 4, the recommendations are very good as it is extremely hard to diagnose epileptiform activity in general. Often EEGs may miss the episode. The authors might consider discussion on novel  wearable EEG microdevices and translation to a family with a child with DS.

The authors could also discuss other (home friendly) technologies that might help diagnose epileptiform activity without having to make a direct observation at a clinic clinic.

In the recommendations section we have added a brief mention of EEG and seizure detection devices. We consider that at the present time there is insufficient evidence for their recommendation in clinical practice in this population, but they could be promising in the not too distant future.

Overall the organization is just right and the topic was well reviewed.

Reviewer 3 Report

This manuscript is a review article describing the increasing incidence of epilepsy amongst adult patients with Down syndrome, particularly a specific type of epilepsy, late-onset myoclonic epilepsy. The great majority of publications on Down syndrome, including those dealing with epilepsy, address infants and children and there are relatively few that focus upon middle-aged and older adults. The authors might consider adding “in adults” at the end of their title of after “Epilepsy” because in infants and children with Down syndrome the risk of epilepsy is low though higher than in typical age-matched controls (except for infantile spasms).

The authors correctly point out well documented neurodegenerative changes of Alzheimer disease occurs in Down syndrome in adolescence and adult life that may be a contributory factor to epilepsy, though Alzheimer disease in adults without chromosomopathies causes progressive dementia but is not particularly epileptogenic. This difference requires more elaboration in the Discussion, even if only speculative because of incomplete published data.

In their discussion of pathogenesis, the authors might cite the now old but landmark neuropathological study by Takashima et al. who demonstrated with Golgi impregnations, that there is a progressive loss postnatally of terminal dendritic ramifications and spines of cortical neurons (Takashima S, Becker LE, Armstrong DL, Chan F: Abnormal neuronal development in the visual cortex of the human fetus and infant with Down’s syndrome: a quantitative and qualitative Golgi study. Brain Res 1981;225:1-21). Though this study examined only infants up to 2 years of age, one might reasonably extrapolate the unique progressive neurodegenerative process beyond age 2 and into adult life. The low prevalence of focal and generalised seizures in infancy and early childhood in Down syndrome may in part be due to a shift in the excitatory/inhibitory synaptic ratio in favour of inhibition, since axo- dendritic synapses on spines are nearly all glutamatergic or excitatory, whereas inhibitory GABAergic synapses are mainly axo-somatic and unaffected (Sarnat HB. Excitatory/inhibitory synaptic ratios help explain epileptogenesis in polymicrogyria and Down syndrome. Pediatr Neurol 2021;116:41-54).

In the discussion of genetics, the authors might consider comparing typical trisomy-21 Down syndrome with the less frequent mosaic variants in the context of risk of epilepsy at various ages.

The manuscript is well written in terms of correct grammar and is well organized into logical sections, including types of seizures, electrophysiological features and pharmacological responses. The authors correctly address many issues of infantile epilepsy in children with Down syndrome. The tables and bullet points are useful. References are appropriate in number and selection, but they might consider adding the couple additional papers cited above.

Author Response

REVIEWER 3:

This manuscript is a review article describing the increasing incidence of epilepsy amongst adult patients with Down syndrome, particularly a specific type of epilepsy, late-onset myoclonic epilepsy. The great majority of publications on Down syndrome, including those dealing with epilepsy, address infants and children and there are relatively few that focus upon middle-aged and older adults. The authors might consider adding “in adults” at the end of their title of after “Epilepsy” because in infants and children with Down syndrome the risk of epilepsy is low though higher than in typical age-matched controls (except for infantile spasms).

Thank you very much for your suggestions.

Regarding the title we are in favor of not adding the word adults. We are aware that epilepsy in children with DS is better characterized than in the adult population. However, we wanted to reflect the bimodal distribution of seizures and part of the work is also focused on reporting the increased risk of seizures in early childhood with respect to the general population. Although it is true that we recognize that we have delved more deeply into epilepsy associated with Alzheimer's disease in Down syndrome. This has been due to the high prevalence of this condition and the need we see as clinical neurologists to provide diagnostic and therapeutic recommendations in this regard.

The authors correctly point out well documented neurodegenerative changes of Alzheimer disease occurs in Down syndrome in adolescence and adult life that may be a contributory factor to epilepsy, though Alzheimer disease in adults without chromosomopathies causes progressive dementia but is not particularly epileptogenic. This difference requires more elaboration in the Discussion, even if only speculative because of incomplete published data.

Increased risk of epilepsy is a phenomenon described in Alzheimer's disease, both sporadic and genetically determined forms. Down syndrome is considered a genetically determined form of Alzheimer's disease. This is not the case in other crosomopathies. There is evidence for a proepileptic role of amyloid in animal models as well as in humans. We believe that it is beyond the scope of this review to expand on other chromosopathies, although it could be very interesting. There are risk factors other than AD in late onset epilepsy, such as the existence of cardiovascular risk factors, which also have a differential presentation in DS (e.g., atherosclerosis in DS is anecdotal). Furthermore, we believe we have explained that the risk of epilepsy in DS is twofold: in early childhood due to anomalies directly associated with the chromosopathy and in late adulthood in relation to the development of symptomatic AD.

In their discussion of pathogenesis, the authors might cite the now old but landmark neuropathological study by Takashima et al. who demonstrated with Golgi impregnations, that there is a progressive loss postnatally of terminal dendritic ramifications and spines of cortical neurons (Takashima S, Becker LE, Armstrong DL, Chan F: Abnormal neuronal development in the visual cortex of the human fetus and infant with Down’s syndrome: a quantitative and qualitative Golgi study. Brain Res 1981;225:1-21). Though this study examined only infants up to 2 years of age, one might reasonably extrapolate the unique progressive neurodegenerative process beyond age 2 and into adult life. The low prevalence of focal and generalised seizures in infancy and early childhood in Down syndrome may in part be due to a shift in the excitatory/inhibitory synaptic ratio in favour of inhibition, since axo- dendritic synapses on spines are nearly all glutamatergic or excitatory, whereas inhibitory GABAergic synapses are mainly axo-somatic and unaffected (Sarnat HB. Excitatory/inhibitory synaptic ratios help explain epileptogenesis in polymicrogyria and Down syndrome. Pediatr Neurol 2021;116:41-54).

We have added the reference of Takashima et al. (reference number 41) following your suggestions. On the other hand, what is concluded in the article by Sarnat et al. is somewhat contradictory with the rest of the literature and even with our own experience in real clinical practice as neurologists. It is undoubtedly an interesting work to consider for the future. The perspective of pathological anatomy studies can always help to better understand the underlying pathophysiology. We will see if the reflected results can be reproducible in other works.

In the discussion of genetics, the authors might consider comparing typical trisomy-21 Down syndrome with the less frequent mosaic variants in the context of risk of epilepsy at various ages.

We have added a comment on future aspects to be evaluated such as genetic risk factors for epilepsy in the section on recommendations and future directions. At this time in the adult population there are no robust publications that allow us to clarify whether the risk of LOMEDS is lower or not in translocations or mosaicism. There is also no evidence on the role of ApoE4 in epileptogenesis. It has been described that ApoE4 in sporadic AD  could increase the risk of epileptic seizures and in fact in the absence of AD in the general population it is believed to be a risk factor for late-onset epilepsy. Given the low frequency of mosaic cases and translocations and lack of robust evidence (as far as we have been able to review) we are not comfortable with stating whether in pediatric age this may have an impact on seizure risk. We believe we could speculate but not assert based on either evidence or our clinical experience.

The manuscript is well written in terms of correct grammar and is well organized into logical sections, including types of seizures, electrophysiological features and pharmacological responses. The authors correctly address many issues of infantile epilepsy in children with Down syndrome. The tables and bullet points are useful. References are appropriate in number and selection, but they might consider adding the couple additional papers cited above.
